# Effectiveness and Safety of Different Treatment Modalities for Patients Older Than 60 Years with Distal Radius Fracture: A Network Meta-Analysis of Clinical Trials

**DOI:** 10.3390/ijerph20043697

**Published:** 2023-02-19

**Authors:** Héctor Gutiérrez-Espinoza, Felipe Araya-Quintanilla, Iván Cuyul-Vásquez, Rodrigo Gutiérrez-Monclus, Sara Reina-Gutiérrez, Iván Cavero-Redondo, Sergio Núñez de Arenas-Arroyo

**Affiliations:** 1Escuela de Fisioterapia, Universidad de las Americas, Quito 170504, Ecuador; 2Escuela de Kinesiología, Facultad de Odontología y Ciencias de la Rehabilitación, Universidad San Sebastián, Santiago 7510157, Chile; 3Faculty of Health, Therapeutic Process Department, Temuco Catholic University, Temuco 4780000, Chile; 4Traumatology Institute of Santiago, Santiago 7501241, Chile; 5Health and Social Research Center, Universidad de Castilla-La Mancha, 16071 Cuenca, Spain; 6Facultad de Ciencias de la Salud, Universidad Autónoma de Chile, Talca 7500912, Chile

**Keywords:** distal radius fracture, surgical intervention, cast immobilization, elderly patients, randomized controlled trial, network meta-analysis

## Abstract

The aim of this study was to compare the clinical effectiveness and complications of different treatment modalities for elderly patients with distal radius fracture (DRF). Methods: We performed a network meta-analysis (NMA) of randomized clinical trials (RCTs). Eight databases were searched. The eligibility criteria for selecting studies were RCTs that compared different treatment modalities (surgical or nonoperative) in patients older than 60 years with displaced or unstable intra-articular and/or extra-articular DRFs. Results: Twenty-three RCTs met the eligibility criteria (2020 patients). For indirect comparisons, the main findings of the NMA were in volar locking plate (VLP) versus cast immobilization, with the mean differences for the patient-rated wrist evaluation (PRWE) questionnaire at −4.45 points (*p* < 0.05) and grip strength at 6.11% (*p* < 0.05). Additionally, VLP showed a lower risk ratio (RR) of minor complications than dorsal plate fixation (RR: 0.02) and bridging external fixation (RR: 0.25). Conversely, VLP and dorsal plate fixation showed higher rates of major complications. Conclusions: Compared with other treatment modalities, VLP showed statistically significant differences for some functional outcomes; however, most differences were not clinically relevant. For complications, although most differences were not statistically significant, VLP was the treatment modality that reported the lowest rate of minor and overall complications but also showed one of the highest rates of major complications in these patients. **PROSPERO Registration:** CRD42022315562.

## 1. Introduction

Distal radius fracture (DRF) is one of the most common types of fractures [1]. DRFs have a bimodal age distribution in the population, with the peak incidence in patients younger than 18 years and the second peak in patients older than 60 years [2]. In elderly patients, DRFs constitute 18% of all fractures [2]. Currently, there are different treatment modalities, such as open reduction and internal fixation with volar or dorsal plates, external fixation (bridging or nonbridging), closed reduction with percutaneous K-wire fixation, closed reduction and cast immobilization, or a combination of these techniques [3].

Generally, treatment depends on the type of DRF, age, activity level, and patient and surgeon preferences [4]. Although surgeon preference and fracture characteristics are the most influential factors in therapeutic decision making, in most cases, surgical treatment is guided by the radiological reduction criteria [5,6]. In the treatment of displaced or unstable DRFs, open reduction and internal fixation with volar locking plates are increasingly being performed because of the general belief that anatomic reduction is positively related to functional outcomes [5,7]. However, the evidence for an association between radiological reduction criteria and functional outcomes is still inconclusive, especially in patients older than 60 years [8,9].

Despite the high incidence of displaced or unstable DRFs, no consensus has been reached on the optimal treatment method in patients older than 60 years [10]. Several systematic reviews with or without meta-analyses have been published on the comparison between surgical and nonoperative treatment in this age group [11,12,13,14,15,16,17,18,19]. These meta-analyses found no statistically significant or clinically relevant differences (the observed differences did not exceed published estimates of the minimally clinically important difference) in functional outcomes between surgical and nonoperative treatment in these patients [13,14,15,16,17,18,19]. Regarding the safety of different treatment modalities in patients with DRF, a recent network meta-analysis (NMA) showed no difference in the risk of complications between nonoperative treatment and any surgical intervention in patients older than 60 years [20].

Although there is a relative increase in studies exploring different treatment modalities for DRFs, all traditional meta-analyses can only compare two treatment options at a time, thereby excluding a large number of studies [20]. To our knowledge, only two NMAs have been published that compared different treatment modalities in patients with DRF; however, these NMAs included a wide age range in patients [3,20]. Accordingly, the aim of this study was to perform an NMA of randomized clinical trials (RCTs) comparing the clinical effectiveness and safety, in terms of functional outcomes and complications, respectively, of different treatment modalities for patients older than 60 years with displaced or unstable DRF. Therefore, the study questions for this NMA were as follows: Which treatment modality showed better functional outcomes? Which treatment modality is associated with the lowest risk of complications in these patients?

## 2. Materials and Methods

### 2.1. Protocol and Registration

This study was conducted and reported according to the PRISMA extension statement for reporting systematic reviews incorporating NMA (PRISMA-NMA) guidelines [21] (PRISMA-NMA Checklist is available in Appendix A) and followed the recommendations of the Cochrane Collaboration Handbook [22]. The registration number in the International Prospective Register of Systematic Reviews (PROSPERO) is CRD42022315562.

### 2.2. Eligibility Criteria

Studies on the effectiveness of surgical interventions and nonoperative treatment for functional outcomes and complications in patients older than 60 years with DRF were considered eligible for inclusion if the following criteria were fulfilled: (1) Population: patients older than 60 years with displaced or unstable intra-articular and/or extra-articular DRFs; (2) Type of intervention: patients treated with any type of surgical intervention for the reduction and/or fixation of a DRF, such as internal fixation with volar or dorsal plates (locking or nonlocking), external fixation (bridging or nonbridging), closed reduction with percutaneous K-wire fixation, or intramedullary fixation; (3) Type of comparison: patients treated with nonoperative treatment (closed reduction and cast immobilization); (4) Types of outcomes: studies that assess clinical effectiveness of wrist and upper limb function with the patient-rated wrist evaluation (PRWE) and the disabilities of the arm, shoulder, and hand (DASH) questionnaires. Secondary functional outcomes included grip strength, wrist range of motion (flexion, extension, pronation, supination), and pain intensity. Studies in which the safety of different treatment modalities was assessed with clinical complications were included. Although there is no standard criteria for classifying complications as major or minor [15,20], our criteria was based on the need for surgery after the initial planned treatment, for surgical treatment interventions, this constitutes reoperation, and for all patients, an unplanned surgery. In this sense, we categorized as minor complications that did not require surgery (i.e., flexor or extensor tendon tenosynovitis, carpal tunnel syndrome, complex regional pain syndrome, delayed fracture, infection, or wrist pain), major complications that required surgery (i.e., implant removal, tendon rupture repair, wrist deformity requiring corrective osteotomy or plate fixation, surgical release for carpal tunnel syndrome, or implant malposition requiring surgical revision), or overall complications; (5) Types of studies: RCTs. Conversely, we excluded studies that included patients with associated fractures of the distal ulna or carpal bones, patients with dislocations of the distal radioulnar joint, patients with vascular injury that required surgical repair, or patients with open fractures.

### 2.3. Electronic Search

We searched the MEDLINE, Central Register of Controlled Trials (CENTRAL), EMBASE, Physiotherapy Evidence Database (PEDro), Latin American and the Caribbean Literature in Health Sciences (LILACS), Cumulative Index to Nursing and Allied Health Literature (CINAHL), SPORTDiscus, and Web of Science (WoS) databases from inception until November 2022. The search strategy for each database is available in Appendix A.

### 2.4. Study Selection

Two reviewers (HG-E and FA-Q) independently reviewed the titles and abstracts of the potentially eligible studies. Subsequently, we obtained the full text. Finally, we added a third reviewer (IC-R) to check for any discrepancy in the selection of a study.

### 2.5. Data Extraction Process

Two authors (IC-V and RG-M) used an ad hoc form to independently extract data on the outcomes for each trial. When data were missing, the researchers contacted the study authors to see the feasibility of obtaining them. If this was not possible, it was decided by consensus to exclude it from the network meta-analysis.

### 2.6. Risk of Bias in Individual Studies

Two researchers (SR-G and SN-A) independently conducted a risk of bias assessment on the RCTs that included using the Cochrane Collaboration’s tool for assessing risk of bias (RoB2) [23]. This tool evaluates the risk of bias according to five domains: randomization process, deviations from intended interventions, missing outcome data, measurement of the outcome, and selection of the reported result. A third reviewer (IC-R) was involved if a consensus could not be reached.

### 2.7. Statistical Methods

Our analysis was performed by the following steps. First, we used a network geometry graph in which the size of the nodes was relative to the number of trials included, and the width of the continuous line connecting nodes was proportional to the number of trials that directly compared two treatment modalities [24]. Network plot graphs were performed with the Confidence In Network Meta-Analysis (CINeMA) [25].

Second, consistency was assessed by checking whether the intervention effects estimated from direct comparisons were consistent with those estimated by indirect comparisons. Confidence was assessed with the CINeMA web application [25].

Third, a standard pairwise meta-analysis was conducted for each direct comparison by using the DerSimonian and Laird random effect method [26]. Pooled estimates of the mean difference (MD) for functional outcomes and risk ratio (RR) with the respective 95% confidence intervals (CIs) for complication rates were calculated. We examined the statistical heterogeneity using the I^2^ statistic; ranging was classified as unimportant (0 to 40%), moderate (30 to 60%), substantial (50 to 90%), or considerable (75 to 100%) [22]. In addition, we considered the corresponding *p* values. Finally, a league table was generated to illustrate these results.

Fourth, we assessed transitivity by checking whether direct comparisons of different treatment modalities included had similar clinical characteristics. Therefore, we evaluated that the patients included in these studies were similar in the baseline distribution of factors such as age, gender, dominant side involved, time of surgery, and time of immobilization post treatment.

Fifth, once we estimated the effectiveness and safety of the different treatment modalities, the surface under the cumulative ranking (SUCRA) was calculated to rank the probability of each type of treatment being the most effective or safe. SUCRA involves assigning a numerical value from 0 to 100 to simplify the classification in the rankogram, with values close to 100 being the best intervention and 0 being the worst [27].

Additionally, subgroup analyses were used to assess the clinical effectiveness and complications of different treatment modalities by the time of follow-up (3, 6, 12, and 24 months) and type of DRF (extra-articular, intra-articular, or both). Finally, Egger’s regression asymmetry test was used to assess publication bias, considering *p* < 0.10 as statistically significant [28]. All analyses were performed using Stata 16.0 (Stata, College Station, TX, USA).

## 3. Results

### 3.1. Study Selection

Electronic searches identified a total of 6097 articles (Figure 1). Finally, 26 studies were included in this NMA [29,30,31,32,33,34,35,36,37,38,39,40,41,42,43,44,45,46,47,48,49,50,51,52,53,54], which included 23 RCTs [29,30,31,32,33,34,36,37,38,39,40,41,42,44,45,46,47,48,49,50,51,53,54] and three articles publishing the long-term follow-up of some of these trials [35,43,52]. The kappa agreement rate between the reviewers was 0.91. Some of these RCTs included patients younger than 60 years; however, the authors of those trials provided their raw data to allow inclusion of their patients older than 60 years in this NMA [30,33,40,42,43,44,47,49,52]. The excluded studies and their reasons are available in Appendix A. Additionally, a summary of the results of direct and indirect comparisons is available in the network forest figures (Appendix A).

### 3.2. Study Characteristics

A summary of the included studies is available in Appendix A. The overall population included 2020 patients (1328 treated with different surgical interventions and 692 with nonoperative treatment). The mean age was 71.1 years (±5.2), and the mean follow-up period was 13.6 months (range, 3 to 36 months). Additionally, the included trials were classified into the following comparisons: (i) volar locking plate versus cast immobilization [29,32,34,35,38,46,48,51,53]; (ii) volar locking plate versus percutaneous K-wire fixation [34,35,37,40,47]; (iii) volar locking plate versus bridging external fixation [34,35,44,49,52]; (iv) percutaneous K-wire fixation versus cast immobilization [31,34,35,54]; (v) volar locking plate versus dorsal plate fixation [42,43]; (vi) bridging external fixation versus percutaneous K-wire fixation [34,35]; (vii) bridging external fixation versus cast immobilization [34,35,36,39,41,50]; (viii) volar locking plate versus intramedullary fixation [33]; and (ix) nonbridging versus bridging external fixation [30].

### 3.3. Risk of Bias Assessment in the Individual Studies

The RoB2 assessment for the included studies is presented in Figure 2 and Figure 3. For the overall risk of bias, 53.8% of the clinical trials were rated as having a low risk of bias [29,30,34,35,38,45,46,47,48,49,51,52,53,54], 34.7% were rated as having some concerns [31,32,33,37,39,42,43,44,50], and 11.5% were rated as having a high risk of bias [36,40,41]. Regarding the randomization process, 53.8% of the clinical trials were rated as having a low risk of bias [29,30,33,36,37,45,46,47,48,49,51,52,53,54]. For deviations from intended interventions, 30.8% of the clinical trials were rated as having a high risk of bias [32,36,37,39,40,41,42,43]. For missing outcome data, 100% of the clinical trials were rated as having a low risk of bias [29,30,31,32,33,34,35,36,37,38,39,40,41,42,43,44,45,46,47,48,49,50,51,52,53,54]. Finally, for the selection of the reported result, 34.7% of the clinical trials were rated as having a low risk of bias [32,34,35,38,44,46,49,51,52].

### 3.4. Network Analyses

The network geometry graphs show the relative amount of evidence available for the different modalities of treatment for functional outcomes such as DASH, PRWE, and grip strength, involving five, two, and seven pairwise direct comparisons, respectively (Figure 4), and on minor, major, and overall clinical complications, involving eight, six, and eight pairwise direct comparisons, respectively (Figure 5). The colors on the graph correspond to the risk of bias in studies included in these comparisons. Additionally, the transitivity assumption was met for all the comparisons by including participants with similar baseline characteristics (age, sex, dominant side involved, time of surgery, and time of immobilization posttreatment). For each comparison, the *p* value was <0.05.

#### 3.4.1. Different Modalities of Treatment on Functional Outcomes

For indirect comparisons, the main summary measure for the analysis was MD. Table 1 shows the MD estimates for the DASH questionnaire. Although no MD was significant, all estimates were in favor of treatment with volar locking plates. Table 2 shows the MD estimates for the PRWE questionnaire. In the NMA, volar locking plate versus cast immobilization (−4.45 points; 95% CI, −8.62 to −0.28) showed a statistically significant difference. Table 3 shows the MD estimates for grip strength. In the NMA, volar locking plates showed statistically significant differences compared to dorsal plate fixation (30%; 95% CI, 16.16 to 43.84), bridging external fixation (6.16%; 95% CI, 1.12 to 11.21), and cast immobilization (6.11%; 95% CI, 2 to 10.21).

#### 3.4.2. Probabilities

For the DASH and PRWE questionnaires, the highest SUCRA values were observed for volar locking plates, as 78.8% and 79%, respectively, and for bridging external fixation, as 57% and 59%, respectively (Table 4). For grip strength, the highest SUCRA values were observed for volar locking plate (97.6%) and percutaneous K-wire fixation (73.9%) (Table 4).

#### 3.4.3. Different Modalities of Treatment on Clinical Complications

For indirect comparisons, the main summary measure for the analysis was the RR. Table 5 shows the RR estimates for minor complications. In the NMA, volar locking plates showed a lower rate of complications than dorsal plate fixation (RR: 0.02; 95% CI, 0.0009 to 0.6) and bridging external fixation (RR: 0.25; 95% CI, 0.07 to 0.93). Table 6 shows the RR estimates for major complications. In the NMA, although no RRs were significant, volar locking plate and dorsal plate fixation showed a tendency toward a higher rate of major complications. Table 7 shows the RR estimates for overall complications. In the NMA, volar locking plate fixation showed a lower rate of complications than dorsal plate fixation (RR: 0.006; 95% CI, 0.0001 to 0.28). Conversely, dorsal plate fixation showed the highest rate of overall complications.

#### 3.4.4. Probabilities

For minor complications, the highest SUCRA values were observed in volar locking plates (84.6%) and intramedullary nails (80.7%) (Table 4). For overall complications, the highest SUCRA values were observed in volar locking plate (84.3%) and cast immobilization (67.5). Conversely, for major complications, the lowest SUCRA values were observed in intramedullary nails (17.8%), dorsal plate fixation (17.9%), and volar locking plates (41.1%) (Table 4).

### 3.5. Subgroup and Publication Bias Analyses

Subgroup analyses were performed to assess the clinical effectiveness and safety of the different treatment modalities according to follow-up time and type of DRF. For direct comparisons in the meta-analysis, compared to cast immobilization, volar locking plates showed statistically significant differences at the 3-month follow-up in grip strength (13.9%; 95% CI, 4.2 to 23.5; *p* = 0.005), DASH (−5.7 points; 95% CI, −8.2 to −3.1; *p* = 0.000), and PRWE questionnaire (−9.2 points; 95% CI, −18.1 to −0.3; *p* = 0.044). At 1 year of follow-up, only grip strength maintained this difference (7.1%; 95% CI, 1.1 to 13.1; *p* = 0.019). Compared to percutaneous K-wire fixation, volar locking plate fixation only showed a statistically significant difference in grip strength at the 3-month follow-up (9.9%; 95% CI, 3.9 to 15.8; *p* = 0.001). A summary of the main findings is available in Appendix A. Additionally, at the 6-month follow-up, the volar locking plate showed a lower risk of minor and overall complications than percutaneous K-wire fixation (RR: 0.12; 95% CI, 0.02 to 0.88; *p* = 0.037) and bridging external fixation (RR: 0.5; 95% CI, 0.29 to 0.86; *p* = 0.025). At 24 months of follow-up, volar locking plate showed a lower risk of minor and overall complications than cast immobilization (RR: 0.69; 95% CI, 0.34 to 1.39; *p* = 0.012). A summary of the main findings is available in Appendix A.

Regarding the type of DRF, for direct comparisons in the meta-analysis, compared to cast immobilization, the volar locking plate showed statistically significant differences in both intra-articular and extra-articular DRFs for grip strength (9.8%; 95% CI, 6 to 13.6; *p* = 0.000), DASH (−2.3 points; 95% CI, −4.2 to −0.4; *p* = 0.016), and PRWE questionnaire (−4.3 points; 95% CI, −7.1 to −1.5; *p* = 0.003). A summary of the main findings is available in Appendix A. Only for intra-articular DRFs did volar locking plates show a lower risk of minor and overall complications than cast immobilization (RR: 0.17; 95% CI, 0.09 to 0.32; *p* = 0.000). Compared to percutaneous K-wire fixation, volar locking plates showed, in both intra-articular and extra-articular fixation, a lower risk of minor complications (RR: 0.3; 95% CI, 0.13 to 0.71; *p* = 0.006) and a higher risk of major complications (RR: 18.38; 95% CI, 3.07 to 110.04; *p* = 0.001). Additionally, for intra-articular DRFs, volar locking plates showed a lower risk of minor and overall complications than bridging external fixation (RR: 0.5; 95% CI, 0.29 to 0.86; *p* = 0.025). A summary of the main findings is available in Appendix A.

Finally, evidence of publication bias was observed through Egger’s test for the following comparisons: volar locking plate versus cast immobilization (*p* = 0.79), volar locking plate versus percutaneous K-wire fixation (*p* = 0.82), volar locking plate versus bridging external fixation (*p* = 0.68), and percutaneous K-wire fixation versus cast immobilization (*p* = 0.85). Additionally, the funnel plot figures are available in Appendix A.

## 4. Discussion

This NMA of RCTs aimed to compare the clinical effectiveness and complications of different treatment modalities for patients older than 60 years with DRF. Despite the increasing number of published RCTs and meta-analyses, no consensus has been reached on the best treatment in elderly patients with displaced or unstable DRF. The main findings of our NMA were that, compared with cast immobilization, volar locking plates showed statistically significant differences in wrist function and grip strength. For DASH, PRWE, and grip strength, the highest SUCRA values were observed in volar locking plates. Conversely, volar locking plate was the treatment modality with the lowest risk of minor and overall complications but it also showed a trend toward a highest rate of major complications. Additionally, subgroup meta-analysis showed that at the 3-month follow-up, volar locking plates were more effective than cast immobilization for upper limb and wrist function and grip strength; at the 1-year follow-up, only grip strength maintained this difference between both treatments. However, these statistically significant differences in the functional outcomes should be interpreted with caution because most differences do not reach the threshold to be considered minimally clinically important [55,56].

Despite the high incidence of displaced or unstable intra-articular and/or extra-articular DRFs and the potential implications of suboptimal management, high-level scientific evidence on the best method of treatment in patients older than 60 years is not available [10,19]. Accordingly, the findings of two NMAs of RCTs performed in all patients with DRF showed controversial results [3,20]. An NMA concluded that open reduction and internal fixation with volar locking plates showed the best results for adults and patients older than 60 years with DRF in terms of short- and long-term functional recovery and the lowest rate of fracture healing complications [3]. Conversely, another NMA showed no clinically important differences in functional outcomes between the different surgical treatments at the 1-year follow-up. For patients older than 60 years, nonoperative treatment may still be the preferred option because there is no reliable evidence showing a decrease in minor or major complications between the different treatment modalities in these patients [20].

Regarding patient-reported outcome measures, most of the studies included in this NMA compared volar locking plate versus cast immobilization [29,32,34,35,38,46,48,51,53]. Consistent with our findings, previous meta-analyses reported no statistically significant or clinically relevant differences in functional outcomes between surgical and nonoperative treatments in patients older than 60 years with DRF [13,14,15,16,17,18,19]. Despite the worse radiographic outcomes associated with closed reduction and cast immobilization, these findings suggest that redisplacement does not affect functional outcomes in these patients [8,19]. Possible explanations could be related to a lower functional demand on the upper limbs, which is thought to be associated with aging, and on the other hand, patient-perceived function or disability is only partially represented by radiological measurements. Accordingly, patient self-evaluation scores will always be subjective, and factors other than those related to the fracture might influence the degree of perceived function or disability [19]. Furthermore, none of the included RCTs reported between-group MD that exceeded the predefined minimally clinically important differences between volar locking plate and nonoperative treatment in functional outcomes [55,56]. However, some of the trials showed small, statistically but not clinically significant differences in favor of patients treated with volar locking plates, which could explain this trend in the magnitude of the effect size in favor of this intervention.

Some surgical modality treatments, such as intramedullary nails, have been used for the stabilization of unstable DFRs, with the aim of reducing soft tissue complications due to a smaller surgical incision while maintaining the benefits of rigid fracture fixation [57,58]. On the other hand, a dorsal plate was used with the aim of allowing the most direct exposure and reconstruction of the joint by a capsular incision but required dissection of the extensor retinaculum and subsequent plate positioning beneath this tendon, which often led to tendonitis or tendon rupture [59]. Despite this, only two trials have compared the effectiveness of these surgical interventions with volar locking plates in patients older than 60 years [33,42]. Further high-quality trials are required to establish the role of these surgical interventions in distal radius fixation.

With the paradigm shift toward surgical treatment of displaced or unstable DRFs with volar locking plates, the overall incidence of residual deformity has decreased [60]. However, this surgical technique has reported variable rates of nerve, tendon, and hardware-related complications [61]. Our findings show that in the short to medium term, volar locking plates showed a lower risk of minor complications than dorsal plate fixation and bridging external fixation. Conversely, compared with other treatment modalities, volar locking plate and dorsal plate fixation showed a trend toward a higher rate of major complications. Despite these findings, there are many discrepancies in the literature on complication rates after a DRF, and we hypothesized that most of the discrepancies are related to the different definitions of a ‘complication’, the stringency with which authors reported complications, and how a major or minor complication is defined in the included studies.

To our knowledge, this NMA of RCTs specifically compared the clinical effectiveness and safety of different treatment modalities in patients older than 60 years with displaced or unstable DRFs. Based on the PRISMA guidelines, the recommendations of the Cochrane Collaboration Handbook, and protocol registration in PROSPERO, this study used a transparent method of assessing and reporting the evidence.

### Limitations

The limitations of our NMA are as follows: (1) there is a moderate to considerable degree of statistical heterogeneity among the included studies, and potential sources of heterogeneity could be variations in the fracture types, different treatment modalities occupied, and cointerventions; (2) methodological limitations, such as a lack of adequate sample size, unclear randomization, inadequately concealed allocation, and lack of blinding of patients and assessors, could overestimate the effect size of the interventions studied; and (3) this study focused only on patients older than 60 years without immediate complications; therefore, the findings do not apply to the younger population, whose functional demands on their hands are greater and might have reported different outcome scores or rates of complications. Finally, our results should be interpreted with caution because high-quality RCTs are needed to establish the clinical effectiveness and safety of different treatment modalities in these patients. Future trials should include methodological strategies to minimize the potential risk of bias, including adequate sample size, randomization process, allocation concealment, and blinding of patients and outcome assessors.

## 5. Conclusions

This NMA showed that compared with other treatment modalities, volar locking plates showed statistically significant differences for some functional outcomes; however, most differences were not clinically relevant. In terms of safety, although most differences were not statistically significant, volar locking plates were the treatment modality that reported the lowest rates of minor and overall complications but also showed one of the highest rates of major complications. This should not be interpreted as a recommendation for surgical management in all patients. The indications for different treatment modalities should be judged based on the balance of risk and benefit in patients older than 60 years with DRF.

## Figures and Tables

**Figure 1 ijerph-20-03697-f001:**
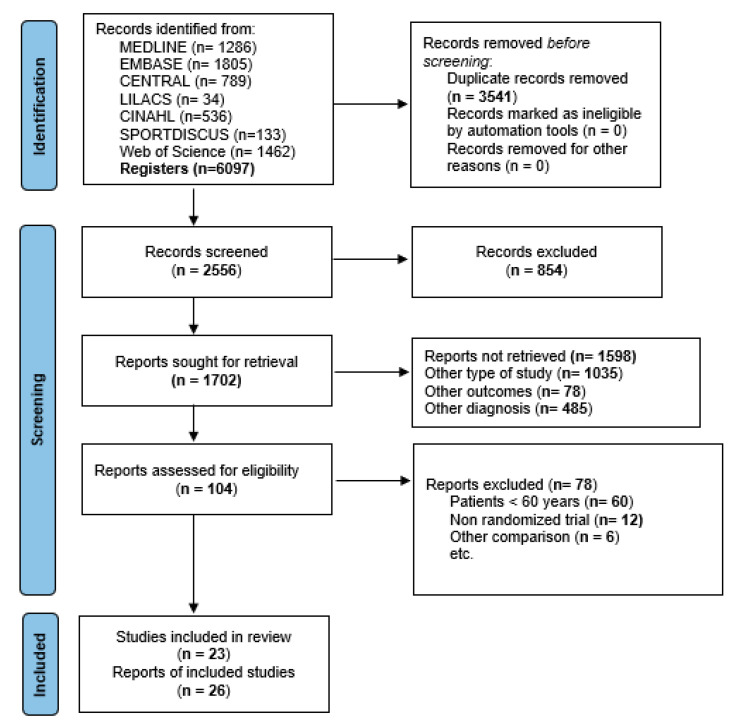
Flow chart diagram.

**Figure 2 ijerph-20-03697-f002:**
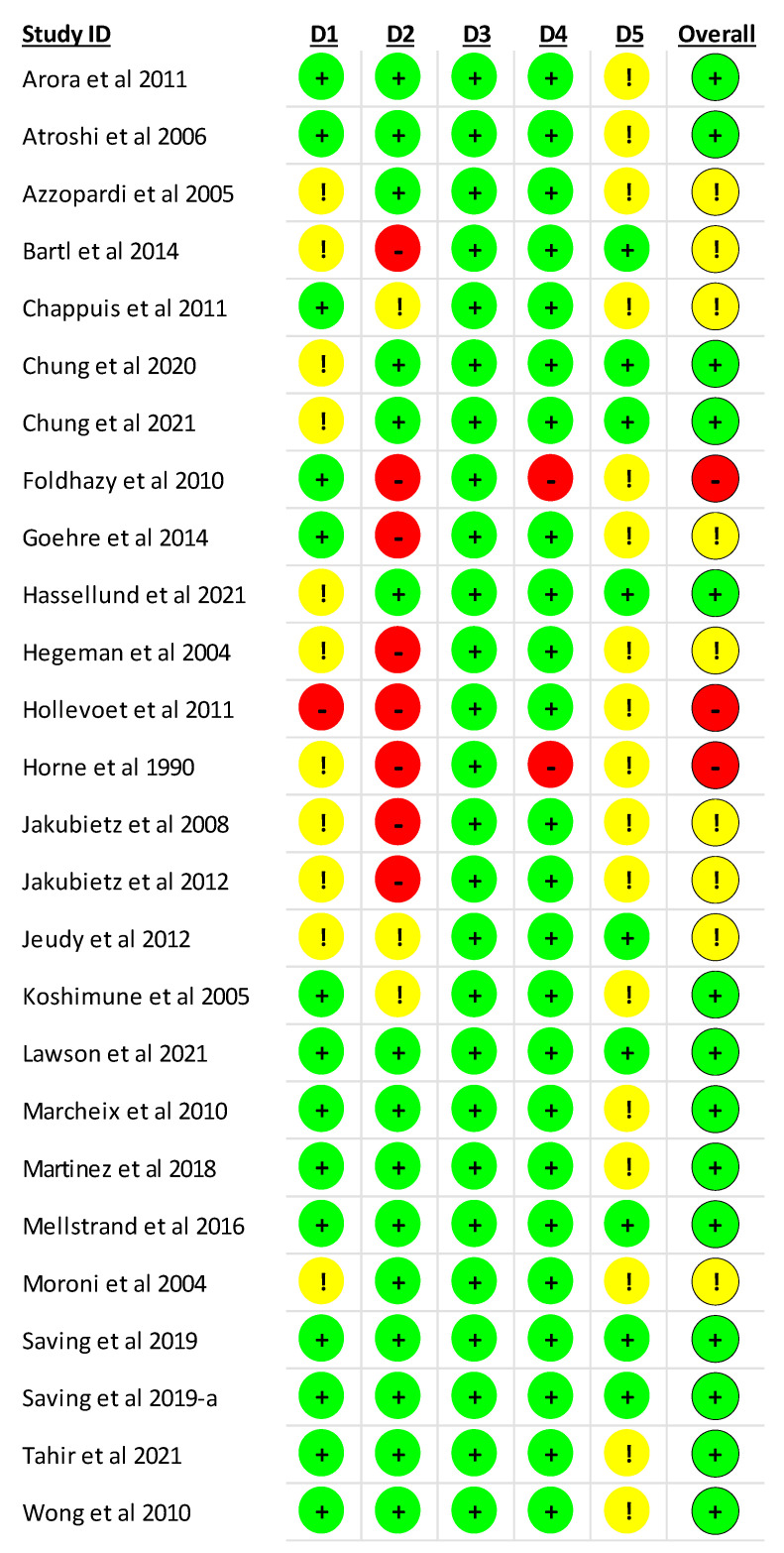
Risk of bias summary: review authors’ judgements about each risk of bias item for each included study [29,30,31,32,33,34,35,36,37,38,39,40,41,42,43,44,45,46,47,48,49,50,51,52,53,54]. Note: “low,” (green ball or “+”) “unclear”, (yellow ball or “!”) and “high” (red ball or “-“) Risk of bias.

**Figure 3 ijerph-20-03697-f003:**
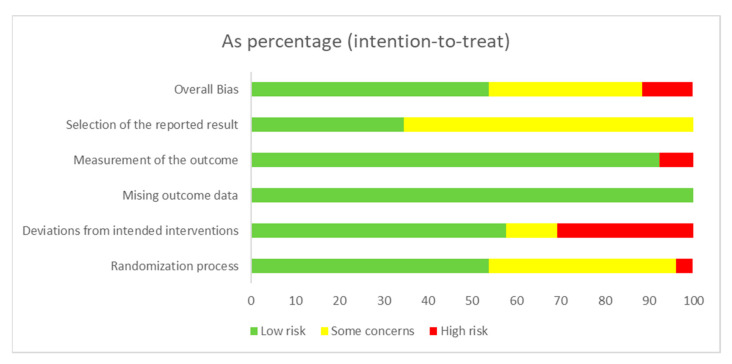
Risk of bias graph: review authors’ judgments about each risk of bias item presented as percentages across all included studies.

**Figure 4 ijerph-20-03697-f004:**
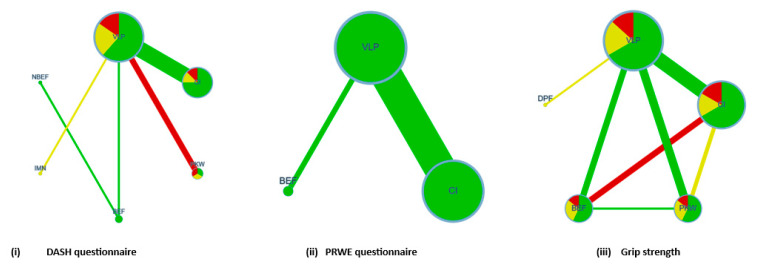
Network geometry for functional outcomes.

**Figure 5 ijerph-20-03697-f005:**
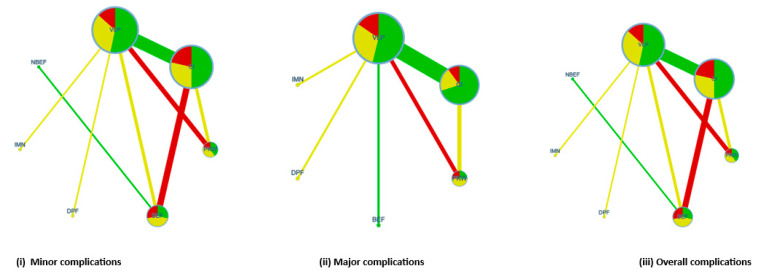
Network geometry for clinical complications.

**Table 1 ijerph-20-03697-t001:** Absolute and relative mean difference estimates (95% CI) on DASH questionnaire. Upper right triangle gives the mean difference from pairwise comparisons (column intervention relative to row) and lower left triangle gives the mean difference from the network meta-analysis (row intervention relative to column).

**Cast Immobilization**	NA	NA	NA	NA	−1.84 (−3.78, 0.10)
−0.7 (−17.12, 15.72)	**Intramedullary Nail**	NA	NA	NA	−2.3 (−16.82, 12.22)
−1 (−7.1, 5.1)	−0.3 (−17.23, 16.63)	**Bridging External Fixation**	NA	4 (−2.48, 10.48)	**−2 (−3.58, −0.42)**
0.64 (−5.78, 7.07)	1.34 (−15.73, 18.42)	1.64 (−6.04, 9.33)	**Percutaneous Kirshner Wire**	NA	−3.8 (−12.19, 4.59)
3 (−8.33, 14.33)	3.7 (−15.74, 23.14)	4 (−5.55, 13.55)	2.36 (−9.9, 14.61)	**Nonbridging External Fixation**	NA
−3 (−6.17, 0.17)	−2.3 (−18.41, 13.81)	−2 (−7.21, 3.21)	−3.64 (−9.29, 2.01)	−6 (−16.88, 4.88)	**Volar Locking Plate**

CI: confidence interval; DASH: Disabilities of the Arm, Shoulder, and Hand questionnaire (scores range from 0 to 100 points, lower scores indicate a better upper limb function); NA: not available. Mean difference in bold: statistically significant. Negative mean differences: the first intervention of the comparison improves upper limb function compared to the second one.

**Table 2 ijerph-20-03697-t002:** Absolute and relative mean difference estimates (95% CI) on PRWE questionnaire. Upper right triangle gives the mean difference from pairwise comparisons (column intervention relative to row) and lower left triangle gives the mean difference from the network meta-analysis (row intervention relative to column).

**Cast Immobilization**	NA	**−4.48 (−8.61, −0.34)**
−3.45 (−12.46, 5.57)	**Bridging External Fixation**	−1 (−3.68, 1.68)
**−4.45 (−8.62, −0.28)**	−1 (−9, 6.99)	**Volar Locking Plate**

CI: confidence interval; PRWE: Patient-Rated Wrist Evaluation questionnaire (scores range from 0 to 100 points, lower scores indicate a better wrist function); NA: not available. Mean difference in bold: statistically significant. Negative mean differences: the first intervention of the comparison improves wrist function compared to the second one.

**Table 3 ijerph-20-03697-t003:** Absolute and relative mean difference estimates (95% CI) on grip strength (*). Upper right triangle gives the mean difference from pairwise comparisons (column intervention relative to row) and lower left triangle gives the mean difference from the network meta-analysis (row intervention relative to column).

**Cast Immobilization**	−2.04 (−8.88, 4.8)	2 (1.82, 2.19)	NA	**7.25 (1.75, 12.76)**
−0.06 (−5.43, 5.32)	**Bridging External Fixation**	**9 (0.36, 17.64)**	NA	4.36 (−4.36, 13.09)
3.69 (−2.45, 9.84)	3.75 (−3.14, 10.64)	**Percutaneous Kirshner Wire**	NA	4 (−4.97, 12.97)
**−23.89 (−38.33, −9.45)**	**−23.83 (−9.1, −38.56)**	**−27.58 (−12.51, −42.65)**	**Dorsal Plate Fixation**	**30 (20.59, 39.41)**
**6.11 (2, 10.21)**	**6.16 (1.12, 11.21)**	2.42 (−3.54, 8.37)	**30 (16.16, 43.84)**	**Volar Locking Plate**

CI: confidence interval; NA: not available; (*): grip strength was measured as the percentage of the injured hand compared with the uninjured hand. Mean difference in bold: statistically significant. Positive mean differences: the first intervention of the comparison improves grip strength compared to the second one.

**Table 4 ijerph-20-03697-t004:** Relative rankings of SUCRA values for functional outcomes and complications of different treatment modalities.

		Volar Locking Plate	Nonbridging External Fixation	Dorsal Plate Fixation	Percutaneous Kirshner Wire	Bridging External Fixation	Intramedullary Nail	Cast Immobilization
DASH	Rank (PrBest)	1st (33.1)	6th (1.3)	-	5th (5.9)	2nd (32.7)	3rd (17.1)	4th (9.9)
	SUCRA	78.8	30.5	-	39.6	57.0	52.1	42.0
	Mean rank	2.1	4.5	-	4.0	3.1	3.4	3.9
PRWE	Rank (PrBest)	1st (58.7)	-	-	-	2nd (40.2)	-	3rd (1.1)
	SUCRA	79.0	-	-	-	59.0	-	12.0
	Mean rank	1.4	-	-	-	1.8	-	2.8
Grip strength	Rank (PrBest)	1st (90.2)	-	5th (0)	2nd (9.8)	4th (0)	-	3rd (0)
	SUCRA	97.6	-	0.0	73.9	30.4	-	48.1
	Mean rank	1.1	-	5.0	2.0	3.8	-	3.1
Minor complications	Rank (PrBest)	1st (57.3)	5th (1.5)	7th (0.5)	4th (3)	6th (0.5)	2nd (28.4)	3rd (8.9)
	SUCRA	84.6	38.5	7.5	44.4	36.8	80.7	57.5
	Mean rank	1.9	4.7	6.5	4.3	4.8	2.2	3.6
Major complications	Rank (PrBest)	5th (3.3)	4th (7.3)	6th (3.2)	2nd (23)	1st (42.5)	7th (0.1)	3rd (20.6)
	SUCRA	41.1	63.5	17.9	68.3	75.9	17.8	65.6
	Mean rank	4.5	3.2	5.9	2.9	2.4	5.9	3.1
Overall complications	Rank (PrBest)	1st (35.3)	5th (5.8)	7th (0.3)	4th (9.3)	6th (1.2)	3rd (13.4)	2nd (34.7)
	SUCRA	84.3	42.4	2.5	48.0	41.2	64.0	67.5
	Mean rank	1.9	4.5	6.9	4.1	4.5	3.2	2.9

**Table 5 ijerph-20-03697-t005:** Absolute and relative RR estimates (95% CI) on minor complications (*). Upper right triangle gives the mean difference from pairwise comparisons (column intervention relative to row) and lower left triangle gives the mean difference from the network meta-analysis (row intervention relative to column).

**Cast Immobilization**	NA	2.49 (0.77, 8.1)	0.52 (0.04, 6.02)	NA	NA	0.48 (0.2, 1.13)
0.28 (0.01, 7.49)	**Intramedullary Nail**	NA	NA	NA	NA	1.5 (0.29, 7.73)
1.87 (0.51, 6.77)	6.7 (0.22, 100)	**Bridging External Fixation**	NA	NA	1.08 (0.68, 1.72)	0.73 (0.37, 1.45)
1.5 (0.28, 8.08)	5.26 (0.16, 100)	0.8 (0.11, 5.88)	**Percutaneous Kirshner Wire**	NA	NA	**0.3 (0.13, 0.71)**
19.48 (0.67, 563.7)	100 (0.75, 1000)	10 (0.31, 1000)	12.5 (0.36, 1000)	**Dorsal Plate Fixation**	NA	**0.15 (0.04, 0.57)**
2.36 (0.11, 51.77)	8.3 (0.09, 1000)	1.27 (0.08, 20)	1.56 (0.05, 50)	0.12 (0.001, 11.1)	**Nonbridging External Fixation**	NA
0.46 (0.19, 1.14)	1.61 (0.07, 33.3)	**0.25 (0.07, 0.93)**	0.3 (0.65, 1.45)	**0.02 (0.0009, 0.6)**	0.2 (0.009, 4.35)	**Volar Locking Plate**

RR: relative risk; CI: confidence interval; NA: not available; (*): number of patients with minor complications. Mean difference in bold: statistically significant. Values less than 1 indicated a protector effect (minor risk) of first intervention compared to the second one.

**Table 6 ijerph-20-03697-t006:** Absolute and relative RR estimates (95% CI) on major complications (*). Upper right triangle gives the mean difference from pairwise comparisons (column intervention relative to row) and lower left triangle gives the mean difference from the network meta-analysis (row intervention relative to column).

**Cast Immobilization**	NA	NA	1.6 (0.2, 12.54)	NA	NA	1.35 (0.78, 2.32)
8.59 (0.36, 204.7)	**Intramedullary Nail**	NA	NA	NA	NA	0.2 (0.01, 3.92)
0.78 (0.31, 2)	0.09 (0.004, 2.33)	**Bridging External Fixation**	NA	NA	NA	1.88 (0.87, 4.1)
0.87 (0.22, 3.46)	0.1 (0.003, 3.03)	1.1 (0.23, 5.26)	**Percutaneous Kirshner Wire**	NA	NA	2.91 (0.49, 17.41)
8.59 (0.36, 204.7)	1 (0.01, 100)	11.1 (0.43, 1000)	10 (0.33, 1000)	**Dorsal Plate Fixation**	NA	0.2 (0.01, 3.92)
0.78 (0.01, 46.23)	0.09 (0.0005, 14.3)	1 (0.02, 50)	0.89 (0.01, 50)	0.09 (0.0005, 14.3)	**Nonbridging External Fixation**	NA
1.5 (0.86, 2.59)	0.17 (0.008, 4)	1.92 (0.84, 4.35)	1.72 (0.44, 6.7)	0.17 (0.008, 4)	1.92 (0.03, 100)	**Volar Locking Plate**

RR: relative risk; CI: confidence interval; NA: not available; (*): number of patients with major complications. Mean difference in bold: statistically significant. Values less than 1 indicated a protector effect (minor risk) of first intervention compared to the second one.

**Table 7 ijerph-20-03697-t007:** Absolute and relative RR estimates (95% CI) on overall complications (*). Upper right triangle gives the mean difference from pairwise comparisons (column intervention relative to row) and lower left triangle gives the mean difference from the network meta-analysis (row intervention relative to column).

**Cast Immobilization**	NA	2.49 (0.77, 8.1)	1 (0.15, 6.85)	NA	NA	0.69 (0.34, 1.39)
1.02 (0.05, 19.27)	**Intramedullary Nail**	NA	NA	NA	NA	0.75 (0.2, 2.8)
2.01 (0.6, 6.71)	1.96 (0.09, 50)	**Bridging External Fixation**	NA	NA	1.08 (0.68, 1.71)	0.8 (0.34, 1.92)
1.69 (0.38, 7.47)	1.67 (0.07, 33.3)	0.84 (0.14, 5)	**Percutaneous Kirshner Wire**	NA	NA	0.49 (0.21, 1.14)
**117.36 (2.3, 5997.8)**	100 (0.96, 1000)	**50 (1.02, 1000)**	**100 (1.15, 1000)**	**Dorsal Plate Fixation**	NA	0.27 (0.07, 1.02)
2.54 (0.14, 45.68)	2.5 (0.04, 100)	1.27 (0.09, 16.7)	1.49 (0.06, 33.3)	0.02 (0.0002, 2.7)	**Nonbridging External Fixation**	NA
0.7 (0.32, 1.55)	0.69 (0.04, 11.1)	0.35 (0.1, 1.19)	0.41 (0.1, 1.67)	**0.006 (0.0001, 0.28)**	0.28 (0.02, 5)	**Volar Locking Plate**

RR: relative risk; CI: confidence interval; NA: not available; (*): number of patients with overall complications. Mean difference in bold: statistically significant. Values less than 1 indicated a protector effect (minor risk) of first intervention compared to the second one.

## Data Availability

Al relevant data are within the manuscript and its Appendix A files.

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
