# Peer review of "Effectiveness and Safety of Different Treatment Modalities for Patients Older Than 60 Years with Distal Radius Fracture: A Network Meta-Analysis of Clinical Trials"

_ijerph, 2023, doi:10.3390/ijerph20043697_

Round 1

Reviewer 1 Report

Dear Author,

This study is a network meta-analysis of RCTs that examined the effectiveness and safety of different treatments for patients older adults with distal radius fractures. The articles were conducted as registered with PROSPERO and reported appropriately under PRISMA-NMA. The introduction and discussion were also clear. I suggest that some modifications be made to the presentation of the results to help the reader understand them.

1. All the figures are low resolution and hard to see. The figure should be understandable on its own, so add a caption for the figure.

2. Add the RoB summary diagram to Fig 2.

3. From the results of the article search (Table S2) and Figure 4, it is possible that this area has not yet matured. Please consider adding a note about the need for further research.

Author Response

Dear Reviewer: Thank you for reviewing our manuscript and giving us the opportunity to improve.

Reviewer 2 Report

  1. Abstract: Only 24 RCTs enrolled in this study, not 27. Please rewrite the result in the abstract based on your combined direct and indirect comparison of the network meta-analysis. 
  2. There were plenty of spelling errors in the manuscript, e.g., “mayor” complications. Please recheck this manuscript carefully.
  3. Please explain the role of the dorsal plate and intramedullary nail in treating DRF in patients aged over 60.
  4. Please explain why to include Koshimune et al. 2005 which compared VLP locking & non-locking. It seemed there was no nonlocking plate included in any outcomes.
  5. Line 97-107 Since only a small portion of studies use DAHS and PRWE as their outcomes. Many outcomes in DASH are based on indirect comparison results, and only three treatments in PRWE. Please explain the appropriateness of using these PROMs as the primary functional outcome. 
  6. Line 103-107 How to define minor and major complications. For example, nonunion is a minor complication, and implant removal is a major complication.
  7. Line 133 Please provide the inter-rater reliability of the two reviewers.
  8. Line 169-174 Please provide the Egger test and the Funnel plot for each outcome in the appendix.
  9. Line 175 Please provide the Network forest result better to understand each outcome's direct and indirect results.
  10. Line 180-182 Please cite those studies that included participants younger than 60 years old, but the trials provided raw data to include patients over 60 years old only for this NMA.
  11. Line 199 Please explain why some include two clinical trials with a strong bias and 11 clinical trials with some concern bias in randomization. Bias in randomization might cause substantial clinical heterogeneity and difficulty in interpreting your result.
  12. Line 342 Please mention the summarized results of the SUCRA of this NMA in the first paragraph of this study.
  13. Line 355 Please point out the reference of MCID of each outcome.
  14. Line 400-402 Please be careful to use “the first” NMA of RCTs comparing different modalities in patients older than 60.

Author Response

(The authors gave the same response as above.)

Round 2

Reviewer 2 Report

Please clarify the inclusion criteria of this study. RCT only or all kinds of comparative clinical trials? For example, Chan 2014 [33] is a retrospective case-control study, which should not be considered an RCT.

The inclusion criteria of some studies are not patients aged over 60, for example, Jakubietz 2008 [43], Jakubietz et al. 2008 [44], and Jeudy et al. 2012 [45]. Please try to review all your included studies again; otherwise, please prevent using “older than 60 years old” from the title. 

Your citation [20] states that CRPS was defined as a major complication but not a minor one in yours. I agree with your definition of classifying complications based on surgery or not; however, it might confuse the readers that some severe complications, such as CRPS or infection, might be underrated.

Heterogeneity is still a problem. The indication of fracture pattern for percutaneous pinning, dorsal plating, and intramedullary nails needs to be applied specifically.

Author Response

ATT: Prof. Dr. Paul B. Tchounwou

Editor-in-Chief

International Journal of Environmental Research and Public Health                                                                                                                     February  2023                              

Dear Editor

Enclosed you will find a revision of our manuscript: “Effectiveness and safety of different treatment modalities for elderly patients with displaced or unstable distal radius fracture: A network meta-analysis of randomized controlled trials” Manuscript ID: ijerph-2194788.

We would like to thank you for giving us the opportunity to revise and improve our manuscript; we also thank the reviewers for their thoughtful and constructive comments.

We have considered all the suggestions and incorporated them into the revised manuscript. All changes are marked with red words. We believe our manuscript is stronger as a result of these modifications. An itemized point-by-point response to the editor and reviewers’ comments is presented below.

We thank you for your time and consideration.
